# Lossy Mode Resonance Generation on Sputtered Aluminum-Doped Zinc Oxide Thin Films Deposited on Multimode Optical Fiber Structures for Sensing Applications in the 1.55 µm Wavelength Range

**DOI:** 10.3390/s19194189

**Published:** 2019-09-27

**Authors:** Patricia Prieto-Cortés, Ricardo I. Álvarez-Tamayo, Manuel García-Méndez, Manuel Durán-Sánchez

**Affiliations:** 1Faculty of Physics and Mathematics, Universidad Autónoma de Nuevo León, San Nicolás de los Garza 66455, Mexico; patricia.prietocts@uanl.edu.mx; 2CONACyT, Universidad Autónoma de Nuevo León, San Nicolás de los Garza 66455, Mexico; rialvarez@conacyt.mx; 3CONACyT, Instituto Nacional de Astrofísica, Óptica y Electrónica, Puebla 72824, Mexico; mduransa@conacyt.mx

**Keywords:** lossy mode resonance, aluminum-doped zinc oxide, optical fiber sensors, multimode fiber, reactive RF magnetron sputtering

## Abstract

In this work, we demonstrated lossy mode resonance (LMR) generation in optical fiber structures based on multimode fibers coated with aluminum-doped zinc oxide (AZO) films. AZO thin films were deposited by using radio frequency magnetron sputtering. In order to exhibit the usefulness of the LMR effect for sensing applications in optical fiber based systems, the deposition conditions of the AZO film coatings were set to obtain the second LMR order within the 1.55 µm wavelength range. An optical transmission configuration setup was used to investigate the LMR effect on fiber structures based on the use of no-core and cladding-removed multimode fibers coated with AZO films synthesized from metallic sputtering targets with different proportions of Zn:Al, 92:8% and 98:2%, at atomic concentrations. The optical and electrical/chemical features of the AZO films were characterized with UV–vis and XPS spectroscopy, respectively. The optical response of the proposed sensing configuration to refractive index (RI) variations was experimentally demonstrated. For the best approach, the sensitivity of wavelength displacement to RI variations on the liquid surrounding media was found to be 1214.7 nm/RIU.

## 1. Introduction

The progress in optical fiber technology has transformed the field of sensing and measurements because of advantages such as compact design, increased user safety due to electrically passive operation, immunity to electromagnetic interference, fast response, high sensitivity, wide sensing range, and remote sensing capabilities, among others. Optical fiber based configurations have demonstrated their reliability for physical, chemical, and biological sensing [1,2]. Currently, the process of thin-film deposition onto optical fibers has been of increasing interest for applications in the realm of optical sensing. Several deposition techniques, combined with the advent of new special fibers and novel materials, have attracted a renewed interest in that area. In this case, optical sensing is achieved by detecting electromagnetic resonances generated by the coupling between a mode guided within the fiber and a mode of the material coating. Depending on the permittivity of the thin-film material, phenomena of surface plasmon resonance (SPR), lossy mode resonance (LMR), or long-range exciton-polariton (LRSEP) are obtained. Although SPR is the phenomenon most studied for sensing applications in fiber systems [3,4,5,6], in recent years, the LMR has generated an increasing interest. For SPR, the film material must possess a negative real permittivity whose magnitude is higher than both its imaginary part and the permittivity of the surrounding medium. Conversely, materials suitable for LMR exhibit the same permittivity conditions for SPR but with the real part of the permittivity being positive [7,8,9,10,11]. Unlike SPR that occurs in thin films of metallic materials such as gold and silver [12,13,14], LMR can be obtained through a wide range of novel materials, such as polymers, metal oxides, and materials with relatively low absorptions [7,15,16,17,18,19,20]. These materials, when deposited onto optical fibers, support multiple modes (lossy modes) near to the cutoff condition, which are guided through the thin film. Besides, unlike SPR where TM or p-polarized light is required for its monitoring, the lossy modes in LMR can be generated with both TE and TM polarized light [21].

Similar to SPR, LMR is generated through attenuated total reflection (ATR) that requires special geometries, of which the Kretschmann–Raether configuration is the most commonly used [10]. However, this bulky prism-based configuration is deemed impractical for many optical sensing applications. In this regard, optical fiber setups offer an attractive alternative for development of LMR-based sensors, since it replaces the use of a coupling prism by the fiber core. In this case, the sensing principle is based on the wavelength interrogation method, where the LMR wavelength is detected as a transmission notch (minimum) in response to the incident light provided by a broadband source [7,8,9,10,11,15,16,17,18,19].

Unlike SPR generation, which exhibits a single resonance maximum, multiple resonances with tunable sensitivities can be generated in LMR. Although the sensitivity of the resonance peaks can be improved by increasing the thickness of the film covering the optical fiber, it also carries a wavelength shift of the LMR toward longer wavelengths [10].

Concurrently, the properties of the film can be adjusted during its fabrication process, together with the proper experimental parameters provided. For this purpose, different methods have been used for metal oxide coating deposition onto optical fiber structures [8,17,19,20]. In this regard, reactive radio frequency (RF) magnetron sputtering has been demonstrated to be a reliable technique for deposition of thin films with high optical quality, accurate thickness control, high surface uniformity, and good sample reproducibility. Moreover, the energy provided to the species during deposition allows a good adherence of films onto flat glass substrates and/or fibers and large deposition areas (from 1 to 10 cm^2^) [22,23,24]. Several transition metal oxides (TMOs), such as indium oxide, indium tin oxide (ITO), titanium dioxide (TiO_2_), and zinc oxide, have been reported as suitable materials for their use as thin film coatings in LMR-based sensors [17,19]. In recent years, different investigations have demonstrated attractive optical properties of AZO, which make it a reliable candidate material for its application in the development of fiber optical systems. Paliwal and John theoretically modeled LMR generation in tapered tip sensors coated with AZO and TiO_2_ layers operating in the visible wavelength range [25]. In a recent work, our research group investigated the nonlinear optical absorption properties of thin films of AZO for their use as saturable absorber material, with the purpose to obtain Q-switched laser pulses in a fiber laser, operating at the 1.55 µm wavelength band [26]. Ozcariz et al. [27], of the research group at the Public University of Navarra, published an experimental work about AZO thin films deposited on fiber structures. These structures were then tested as LMR refractometers operating in the visible wavelength range and near infrared (around 1300 nm).

As a transition metal oxide that supports multiple modes, AZO exhibits favorable optical properties for LMR generation. Moreover, AZO is a nontoxic material with high availability and low cost for large-area applications. When ZnO grown as a thin film is doped with Al (from 1% to 10% in atomic concentration into the major ZnO matrix), the extra electrons supplied for the aluminum induce changes in the optoelectronic properties of the whole material [28]. In this sense, the detailed study of the parameters used during the deposition process, together with the detailed characterization about modifications of properties at the electronical/optical level, is a subject of interest in order to implement AZO films with tunable properties for their use in fiber-based LMR sensors.

Moreover, LMR fiber sensors have been mainly implemented in transmission configuration [3,8,17,19,20]. In this case, different optical fiber structures, such as tapered fibers [15,25,29], D-type polished optical fibers [12,15], and cladding-removed multimode optical fibers (CRMOFs) [15], among others, have been used in combination with the material coating in order to sense the intended target parameter. The aforementioned structures require the use of techniques such as micromachining and chemical attack, whose processes are usually inaccurate, which is detrimental to the precise study of the generalized characteristics of the sensor, as well as its implementation and repeatability. In this regard, the use of a multimode no-core optical fiber (NCF) allows easy construction of reliable and repeatable fiber structures suitable for use in development of LMR fiber sensors.

Furthermore, the 1.55 µm waveband encompasses remarkable advantages for optical fiber systems such as low transmission losses and achievable implementation with current technology and equipment.

Then, the development of LMR sensors in the 1.55 µm wavelength region is attractive because it is possible to implement optical devices that are compatible with fiber systems. In addition, the optical/electrical properties of the coating material significantly vary depending on the setting of the material composition and the deposition process parameters, which allow the tuning of the LMR wavelength for its generation in the 1.55 µm wavelength band. Then, a detailed analysis about the optical and electrical properties of AZO films covering the fiber, as well as the comparison of different fiber structures used, can be useful in order to determine the reliability, sensitivity, and repeatability of the optical sensor.

In this paper, we present an experimental study on LMR generation in optical fiber structures coated with AZO thin films deposited by reactive RF magnetron sputtering. In our investigation, based on previously reported works on AZO-coated fiber structures, we generated LMR with second order resonance, suitable for use in the development of fiber sensors in the 1.55 µm wavelength region, which was possible to achieve because of the deposition process parameters of the AZO films. In this regard, the XPS, UV–vis spectroscopy, and Filmetric characterizations of the deposited thin films are shown in detail. In addition, thin film depositions with different Zn and Al metal precursors, in proportions of 98:2 and 92:8 wt%, are discussed for NCF- and CRMOF-based fiber structures. The operation characteristics of the implemented LMR sensors in optical transmission setup are discussed. The optical response of the proposed sensing configuration to refractive index (RI) variations in solutions of isopropyl alcohol (IPA)/glycerin mixtures is presented. The transmitted optical spectrum of the fiber sensor exhibits a wavelength shift of the resonance absorption peak toward longer wavelengths, as the surrounding RI is increased. For the best fiber sensing structure, the RI sensitivity was found to be 1214.7 nm/RIU.

## 2. Thin Film Coated Fiber Structures for LMR-Based Sensors

### 2.1. Single-Mode–Multimode–Single-Mode Fiber Structures

The single-mode–multimode–single-mode (SMS) structure used in our experiments was formed by a multimode fiber (MMF) segment fusion spliced between two single-mode fiber (SMF) segments, as is shown in the schematic of Figure 1a. Two different MMF fibers were used. In the first configuration, an MMF (Thorlabs 105/125) whose silica cladding was etched by immersion of the fiber in hydrofluoric acid (HF) solution with a concentration of 40% was used. In order to monitor the cladding that was removed as it was attacked with the HF solution, we used the transmission configuration reported by El-Sherif in Ref. [30]. By using a 978 nm multimode laser as the input source for the aforementioned configuration, the output power drop indicated when the cladding was removed. In our case, the power dropped at an immersion time of ~7 min. In the second configuration, a homemade NCF replaced the CRMOF in the SMS fiber structure. The NCF had a silica cladding with a diameter of 125 µm, noncircularity of less of 3%, refraction index of 1.446 at 1550 nm, and return losses >50 dB. For both fiber structures, the fiber length fused between the SMFs was 4 cm. Use of the NCF avoided the use of cladding removal techniques and minimized the losses and interference modulations produced by mismatched diameter fiber splicing.

### 2.2. LMR Generation in Thin Film Coated Optical Fibers.

Without a thin film coating, the SMS structure acts as a narrow band-pass wavelength filter based on the multimode interference effect [31]. As it is shown in the schematic of Figure 1a, the light from the input SMF is propagated within the MMF through different modes. The coupling coefficient of the excited mode interference produces periodic self-images at different MMF fiber lengths, in which a high transmission peaks at a corresponding wavelength is reached. In order to obtain a field distribution at the end of the MMF as an image of the input field, the length of the MMF for a central wavelength where the maximum multimode interference is obtained as follows [31].
(1)$$\lambda_{c}=p\frac{n_{MMF}D_{MMF}^{2}}{L}$$
where *p* is the self-image number, *n_MMF_* is the effective refractive index of the MMF, *D_MMF_* is the diameter of the MMF corresponding to the effective width of the fundamental mode, and *L* is the MMF length. However, when the MMF surface is deposited with a material film, the self-image spectrum shifts toward longer wavelengths depending on the thickness of the coating. At the same time, the self-image band approaches a wavelength region in which the self-image effect fades [32], which is caused by electromagnetic resonance [33].

Optical fiber sensors based on electromagnetic resonance responses are based on the dielectric properties of the material thin films deposited onto the fiber surface. The resonance phenomenon (SPR, LMR, or LRSEP) related to a specific material coating depends on its electrical permittivity. LMR generation occurs when the real part of the permittivity of the coating material is positive and greater in absolute value than its imaginary part as well as the permittivity of the material surrounding the coating. Then, the relation between the permittivity of the material (ε) and its complex RI (*N = n + jk*) establishes the conditions for the resonance phenomenon:(2)$$\varepsilon =\varepsilon_{R}+j\varepsilon_{I}={\left(n+jk\right)}^{2}=n^{2}-k^{2}+j2nk$$
where $\varepsilon_{R}=n^{2}-k^{2}$ and $\varepsilon_{I}=j2nk$. Then, for LMR generation of a material coating deposited onto an optical fiber with permittivity $\varepsilon_{sur}=\varepsilon_{R_{sur}}+\varepsilon_{I_{sur}}$, the permittivity conditions of $\varepsilon_{R}>\varepsilon_{I}$ (refractive index condition of $\left|n_{2}\right|>\left|k_{2}\right|$) and $\varepsilon_{R}>\varepsilon_{R_{sur}}$ must be satisfied.

The LMR phenomenon is attributed to the mode coupling between a fiber-guided mode and a coating-guided lossy mode with considerable phase coincidence, which occurs when both modes are propagated though the fiber near to the cutoff condition. From this condition, the light mode is guided through the material coating depending of the wavelength of the excitation light source and the coating thickness.

The use of a fiber structure reduces the complexity of a conventional Kretschmann–Raether ATR configuration commonly used in bulky optical systems for this purpose [10]. In an SMS fiber structure configuration, the reflectivity as a function of the input light wavelength and the incidence angle at the fiber/coating interface is obtained by a Kretschmann-based ATR method. Since the input light is not polarized, the reflectivity depends on the reflected power for both TE and TM polarization modes:(3)$$R\left(\theta ,\lambda \right)=\frac{R_{TM}\left(\theta ,\lambda \right)+R_{TE}\left(\theta ,\lambda \right)}{2}$$

Multiple reflections occur in the MMF core because of the guided support modes. In this case, total internal reflection (TIR) at the fiber/coating interface along the deposited MMF is obtained by incident light modes with angles greater than the critical angle, as is shown in Figure 1b. Then, coupling between the TIR-generated evanescent wave and the supported modes by the thin film material is obtained [7,33,34]. The number of reflections at the fiber/coating interface as a function of the angle of incidence *N_R_(θ) = L/d*tan*(θ)* (neglecting the skewness angle) depends on the length (*L*) of the coated region and the fiber diameter (*d*). Considering a nonremote sensing case, due to the short dimensions of the fiber, the transmitted power is given by [35].
(4)$$T\left(\lambda \right)=\frac{\int_{\theta_{c}}^{90{}^{\circ}}p\left(\theta \right)R^{N_{R}\left(\theta \right)}\left(\theta ,\lambda \right)d\theta }{\int_{\theta_{c}}^{90{}^{\circ}}p\left(\theta \right)d\theta }$$
where *p(θ)* is the input light source power distribution, and *R^N^_R_^(θ)^* is the reflected light at the fiber-coating interface. *θ_c_* is the critical angle in which the TIR transmission is obtained, expressed by:(5)$$\theta_{c}=arcsen\left(\frac{n_{clad}}{n_{core}}\right)$$
where *n_clad_* and *n_core_* are the refractive indexes of the cladding and the core, respectively.

In our case, the broadband LED-based light source used can be modeled as Gaussian field distribution:(6)$$p\left(\theta \right)\propto e^{\left[-\frac{{\left(\theta -\pi /2\right)}^{2}}{2W^{2}}\right]}$$
where *W* is the width of the Gaussian function. Thus, when light is absorbed at the core/coating interface, a transmission spectrum that exhibits multiple absorption peaks at wavelengths located at maximum EM resonances is obtained.

In this case, AZO is a novel TMO material whose properties make it feasible for use as LMR supporting material [36]. Depending on the deposition techniques and parameters, the electrical/optical properties of the deposited film can significantly vary. For permittivity modeling of AZO, the Drude–Lorentz model is appropriate for the infrared wavelength region:(7)$$\frac{\varepsilon }{\varepsilon_{0}}=1-\frac{\omega_{p}^{2}}{\omega^{2}-j\omega \gamma }$$
where *ε_0_* is the permittivity of the vacuum, *ω_p_* is the plasma frequency of the material, and γ is the damping constant for oscillator. As it is expected in LMR materials, the thickness of the thin film is the key parameter. When the thickness of the thin film is increased, the sensitivity of the LMR orders is improved as the resonance peak shifts toward longer wavelengths [27]. In addition, the sensitivity of LMR to RI changes in the surrounding medium and decreases with the increase of coating thickness [7]. Also, the extra electrons supplied for Al doping concentrations in AZO thin films induce changes in the optoelectronic properties of the whole material [28]. Then, the parameters of the AZO thin film deposition process and the characterization of properties at the electrical/optical level allow the accurate thickness and doping control required in AZO coatings onto optical fibers for use in the development of LMR-based optical sensors.

## 3. Deposition of AZO Films on Fibers and Coating Characterization

The AZO films were deposited on the fiber structures with the technique of RF magnetron sputtering. Metallic circular targets (1′′ diameter, 1/8′′ thickness) of Zn:Al alloys were used: one target of Zn:Al (98%:2%) and another one of Zn:Al (92%:8%) atomic concentrations at 99.99% atomic purity. We used commercially available targets provided by Feldco International. Doping concentrations of 2% and 8% were chosen, since this was the range reported for a useful Al-doped ZnO film with almost metallic-conductive properties. Because of the sputtering process, the initial proportion of Al into the ZnAl alloy target can be altered when the AZO film is grown. This is the other reason why two different targets (with two Al-doping proportions) were chosen too. A doping of about 1% to 2%, at least, of Al into the ZnO film must be achieved. From previous experience, this range of Al doping, together with the right experimental parameters, can assure that Al is desirably doped on the AZO film.

Optical fibers were mounted on glass substrates. As the sputtering process is energetic enough, the deposited film covered the external side of the optical fiber. The internal diameter was left untouched. Previous experiments were performed to assure this. The substrates were also characterized in order to analyze the optical-electrical properties of films covering the fibers. The fibers/substrates were mounted in a holder, which was located parallel to the target, at a distance of 5 cm. All depositions were performed inside a bell-jar glass chamber with a base pressure of 5.5 × 10^−5^ Torr. The sputtering process took place at a working pressure of 10 mTorr. A mixture of high-purity gases (99.99%), Ar and O, were employed in the reaction. Flow rates of 20 and 1 cm^3^/min for Ar and O, respectively, were used. Flow of gases was controlled with electronic mass flowmeters. Each mass flowmeter was calibrated for the specific gas passing through it. Plasma was generated by applying an RF power of 30 watts. Prior to deposition, the target was pre-sputtered for 5 min to remove any contamination. A mechanical movable shutter was located between the target and the substrate. After the pre-sputtering process was completed, the shutter was retired, and the deposition process took place.

The deposition process was controlled by monitoring the deposition rate, which was measured through a quartz crystal sensor interfaced with an external computer. A quartz monitor can measure both evaporation rate and thickness as a function of deposition time. The deposition time was 10 min. No temperature was applied in the process. Although the evaporation rate can remain the same between depositions, targets are eroded with time. For this reason, the thickness that appeared in the monitor (in thickness mode during deposition) can lead to mistakes. Thus, in order to avoid inaccuracies, the thickness of deposited films was measured ex situ by Filmetrics equipment in reflectance mode. Measurements were conducted in regions of the substrate very close to the fiber. Information about the attained coatings and type of fibers is given in Table 1.

The composition of films was analyzed by XPS (K-ALPHA by Thermo Fisher Scientific, Waltham, MA, USA) using a monochromatic AlKα source of 1486.68 eV. The film surface was sputtered in situ with Ar^+^ ions in order remove surface contamination adsorbed on samples. Software from equipment was used to calculate the atomic concentration of elements as well as the chemical state. The binding energies were calibrated with the C1s peak at 284.5 eV. The procedure employed was to first get a survey spectrum, followed by windows of high-resolution for Zn2p, O1s, and Al2p transition spectra, before and after sputtering. Optical properties were characterized from transmittance measurements using a UV–vis–NIR spectrophotometer (double-beam Jasco V-770) in the wavelength range of 300 to 2500 nm.

Figure 2 displays the XPS measurements (a) survey spectrum, (b) Zn2p, (c) O1s, and (d) Al2p high resolution window spectra corresponding to sample 1 after sputtering. In the survey spectrum, all elements detected were labeled together with the Zn LMM Auger signal.

In Figure 2b, the Zn2p core level was conformed of two sublevels: 2p_1/2_ and 2p_3/2_ (due to spin-orbit splitting). The binding energy (BE) of Zn2p_3/2_ was located at 1021.8 eV. As the BE of zinc exhibits small energy shifts, chemical state differentiation with only the Zn2p transition is difficult. As the LMM Auger peak has a larger shift with the chemical state, it was used for chemical identification. The reported kinetic energy of the Auger LMM for metallic Zn is at 992 eV, while the one for oxidized zinc is at 988 eV [37]. For sample 1, the KE of the Zn LMM Auger transition was detected at 988.6 eV (489 eV in BE). The same value was obtained for samples 2 and 3. No metallic Auger transition was detected in the survey spectra.

In Figure 2c, the O1s spectrum, the convoluted signal is divided into three components: LBE attributed to the Zn–O bond of crystal lattice oxygen, at a BE range of about 529.7–530.5 eV; IBE, related to O^2−^ ions in oxygen-deficient regions within the matrix of ZnO, with BE values in the range of about 530.8–531.6 eV; and the HBE part, attributed to oxygen in adsorbed water or carbonates, with the BE in the range of about 532–534 eV. For sample 1, the LBE was located at 530.08 eV, and for samples 2 and 3, at 530.01 and 529.98 eV, respectively. For the three coatings, LBE was the highest in intensity. The relative percentages of LBE, extracted from deconvoluted curves, were about 81.0%, 83.8%, and 83.4% for samples 1, 2, and 3, respectively.

In Figure 2d, when Al was in the metallic state (Al–Al bond), BE was about 72.6 eV. When Al was oxidized, BE shifted from 72.6 eV to the highest values [38,39,40]. The BE of Al2p of sample 1 was located at 73.6 eV. For samples 2 and 3, BEs were at 73.50 and 73.58 eV, respectively.

Taking in to account the Zn2p_3/2_ transition (oxidized zinc), the O1s LBE component (oxygen bound to zinc), and the Al2p signal, the empirical formula was obtained from calculations of atomic concentration. The results are included in Table 2.

Coatings were not stoichiometrically balanced, which can be attributed to the deposition conditions such as the low RF power and no temperature applied. In this case, films were not completely crystalline. The concentration of Al in sample 1 was about 9%, slightly higher than the Al in samples 2 and 3. This difference stems from the type of target used in the deposition process.

The optical transmittance of AZO films is shown in Figure 3. The films showed a transparency of about 90% on average in the range of 500 to 2500 nm. The region of λ < ~500 nm was the onset of electronic transitions, where the process of optical absorption took place. At the onset of the absorption edge, the optical band gap (*E_g_*) can be obtained through Tauc curves. Details about this method have been described [41]. In these curves, a graph of (αE)^2^ (eV^2^cm^−2^) versus E (eV) is constructed. Then, the band gap is found by extrapolating the linear part of the curve at α = 0. Results are included in Table 2.

Concerning the values of *E_g_* found for the samples, no meaningful difference existed among them. In an earlier work, values of *E_g_* ~3.2 eV were obtained for RF-grown ZnO films using the same equipment used to grow the current AZO films [41]. For AZO films prepared with RF and DC sputtering techniques, authors report that values of *E_g_* showed variations dependent on the deposition conditions. Authors of Ref. [42] report RF-grown AZO thin films that were deposited at 50 and 70 W and showed optical bandgap energies of ~3.67 eV, with a transmittance of ~90% in the visible region. Additionally, authors in Ref. [28] for DC-sputtered AZO thin films found values from 3.15 to 2.97 eV with an increase in the target power from 100 to 250 W. Thus, the values of *E_g_* found for our films fell within the reported values of sputtered-grown AZO films.

## 4. Results and Discussion

The schematic of the experimental setup for RI sensing is shown in Figure 4. The AZO-coated fiber structure was placed in contact with different concentrations of IPA/glycerin solutions by using a homemade mechanical device, where the fiber structure was placed and fixed straight between holders. A fiber-coupled LED source with light emission in the wavelength range from 1400 to 1700 nm was used as the input signal. The transmitted optical spectrum of the light from the input source passing through the coated fiber structure was measured by using an optical spectrum analyzer (OSA, Anritsu MS9710B) with a resolution of 0.07 nm.

Figure 5 shows the transmission spectra of the fiber samples for different concentrations of IPA/glycerin solution. The spectra show the second-order LMR in the 1.55 µm wavelength range. The LMR absorption peak shifted toward longer wavelengths as the glycerin concentration increased, which was due to the increase of the RI of the surrounding media. Figure 5a–c shows the transmitted spectrum for fiber structures of samples 1, 2, and 3, respectively. The RI of the IPA/glycerin solution for each concentration is also indicated. As can be observed, depending on the fiber structure and the deposition conditions, each resonance exhibited different transmission notch widths, depths, and sensitivities.

Based on the experimental results obtained in Figure 5, the sensitivities of each fiber structure, with wavelength shifts of the resonance transmission notch as a function of the RI variations of the surrounding media, are shown in Figure 6. The wavelength displacement of the LMR peak, due to RI variations of different concentrations of IPA/glycerin solutions, can be linearly fitted. Then, the sensitivity of each fiber structure was obtained. As it can be observed, the fiber structure based on the use of NCF and AZO coatings with 8% aluminum doping exhibited a higher sensitivity of 1214.7 nm/RIU, whereas the lowest sensitivity of 937.8 nm/RIU was obtained for the CRMOF-based fiber structure with 2% Al-doped AZO coating.

Sample 1 and sample 2 were constructed with the same fiber structure. In this case, because the film deposited in sample 2 was thicker than in sample 1, the central wavelength of the resonance peak of sample 2 was expected to be longer than in sample 1; however, the contrary was observed. In this sense, we attributed this to the fact that the increase in the concentration of Al doping contributed more significantly to the displacement of the resonance peak towards longer wavelengths than the thickness of the film as well to the sensitivity of the coated fiber structure. Moreover, samples 2 and 3 were AZO-coated with the same deposition conditions (simultaneously); then, the difference in sensitivity was attributed to the diameter difference of the multimodal fibers (125 µm for NCF of samples 2 and ~105 µm for the CRMOF of sample 3), so that the fiber with a larger diameter was more sensitive to RI changes in the surrounding media. Additionally, the second-order LMR notch showed a significantly narrower spectral width for the fiber structure of sample 3, whose multimodal fiber diameter was shorter than samples 1 and 2.

## 5. Conclusions

LMR generation in optical fiber structures, coated with AZO thin films deposited by reactive RF magnetron sputtering in the 1.55 µm wavelength range, was experimentally demonstrated. The generation of LMR using fiber structures coated with AZO films with varying thicknesses and contents of Al was evaluated. The wavelength displacement of the second-order LMR resonance notch to RI variations of each fiber structure, submerged inside solutions of isopropyl alcohol (IPA) with glycerin, was discussed for the purposes of sensing applications. Thus, wavelength shifts to RI variation sensitivities of 1214.7, 1002.1, and 937.8 nm/RIU, for samples 1, 2, and 3, respectively, were obtained.

## Figures and Tables

**Figure 1 sensors-19-04189-f001:**
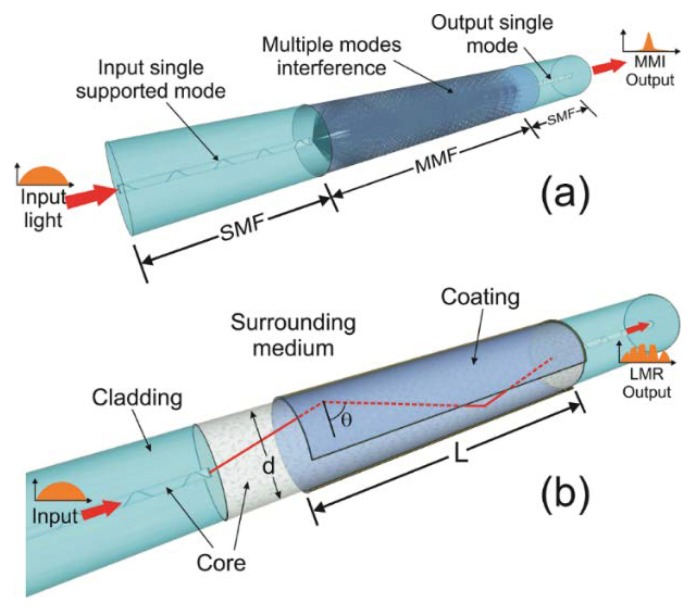
Schematic of the single-mode–multimode–single-mode (SMS) fiber structure: (**a**) without coating (multimode interference effect), and (**b**) with a sensitive region *L* from the thin film coating (lossy mode resonance (LMR) phenomenon).

**Figure 2 sensors-19-04189-f002:**
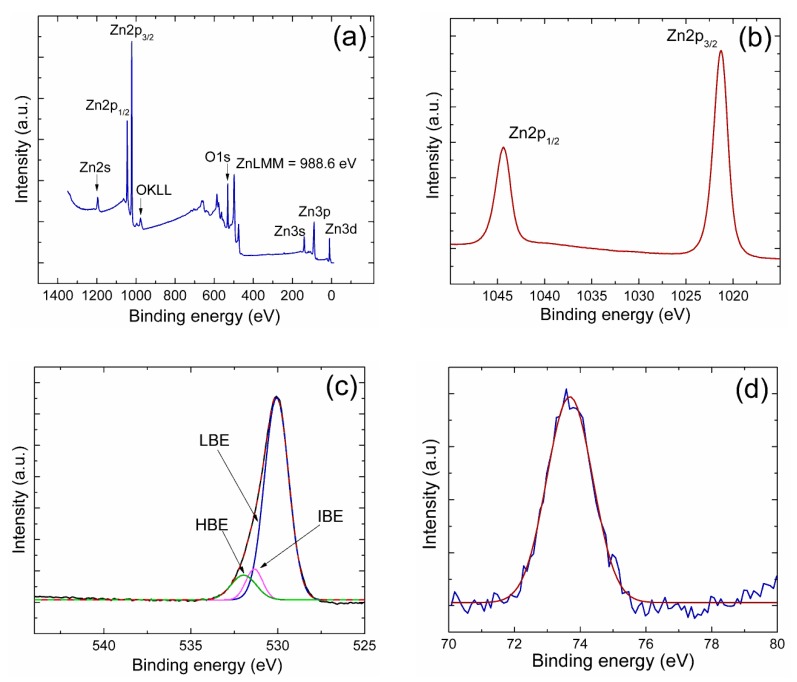
XPS measurements corresponding to the coating of sample 1 after sputtering: (**a**) survey spectrum; (**b**) Zn2p, (**c**) O1s, and (**d**) Al2p high resolution window spectra.

**Figure 3 sensors-19-04189-f003:**
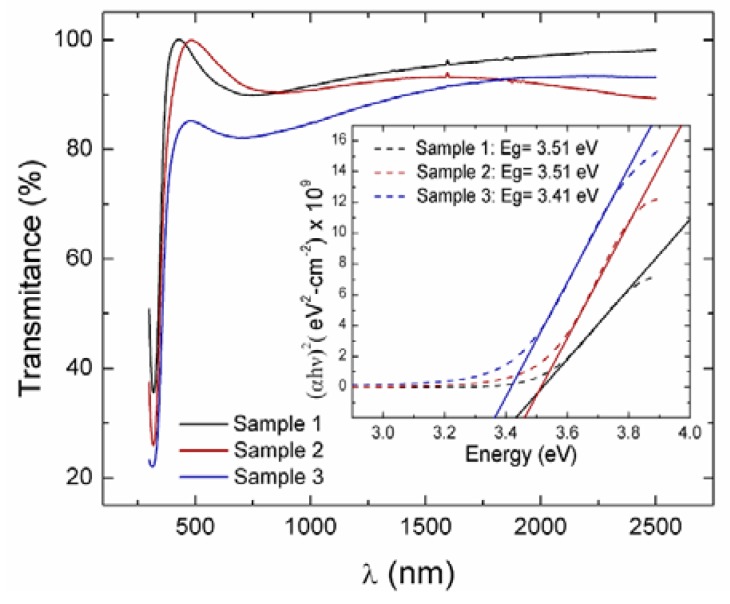
Transmittance spectra of AZO films. Inset: Tauc curves.

**Figure 4 sensors-19-04189-f004:**
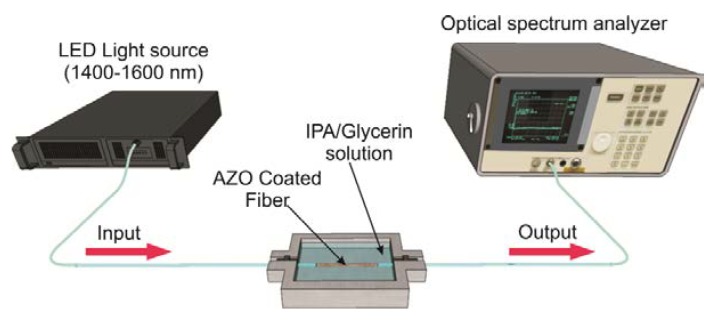
Experimental setup of the refractive index sensing optical fiber system. IPA, isopropyl alcohol.

**Figure 5 sensors-19-04189-f005:**
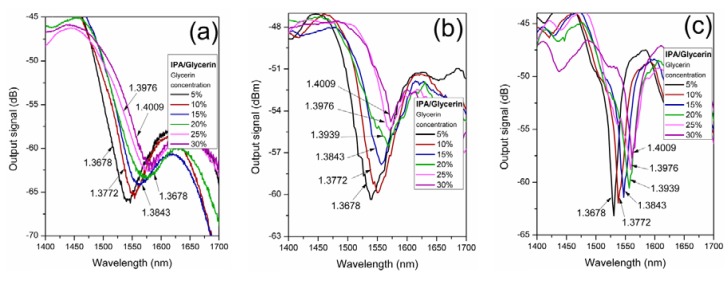
Transmitted output spectra of the coated fiber structure responses to IPA/glycerin concentration variations for: (**a**) sample 1, (**b**) sample 2, and (**c**) sample 3.

**Figure 6 sensors-19-04189-f006:**
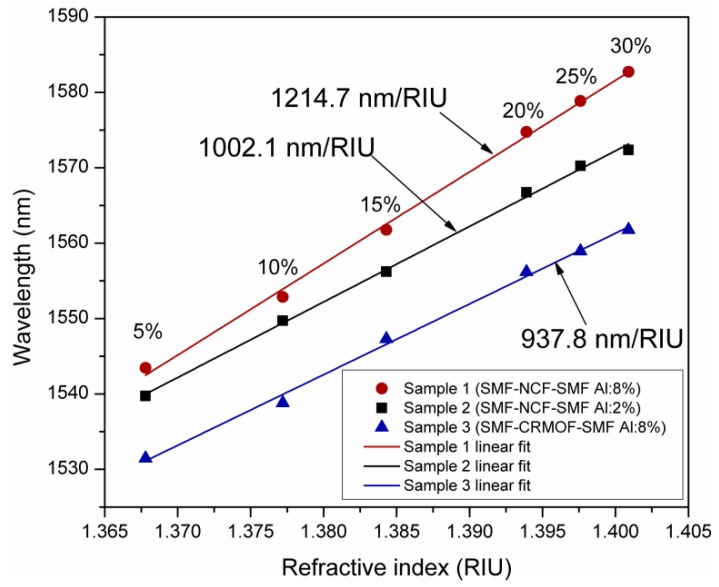
Sensitivity of the coated fiber structures to RI variations of the IPA/glycerin solutions.

**Table 1 sensors-19-04189-t001:** Features of deposited aluminum-doped zinc oxide (AZO) coatings and type of fiber.

Sample	Target (% at. Conc.)	Fiber Structure	Thickness (nm)
1	Zn:Al, 92:8%	SMF–NCF–SMF	110
2	Zn:Al, 98:2%	SMF–NCF–SMF	160
3	Zn:Al, 98:2%	SMF–CRMOF–SMF	150

SMF, single-mode fiber; NCF, no-core optical fiber; CRMOF, cladding-removed multimode optical fiber.

**Table 2 sensors-19-04189-t002:** Empirical formulae and optical band gap for AZO films.

Sample	Empirical Formulae	E_g_ (eV)
1	Zn_0.30_:O_0.59_:Al_0.09_	3.51
2	Zn_0.32_:O_0.60_:Al_0.07_	3.51
3	Zn_0.32_:O_0.60_:Al_0.07_	3.41

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
