# Peer review of "Lossy Mode Resonance Generation on Sputtered Aluminum-Doped Zinc Oxide Thin Films Deposited on Multimode Optical Fiber Structures for Sensing Applications in the 1.55 µm Wavelength Range"

_sensors, 2019, doi:10.3390/s19194189_

Round 1
Reviewer 1 Report
Response:
In this article the authors claim in an optical fiber sensors using an optical-fiber structures based on multimode fibers and NCF coated with an aluminum-doped zinc oxide (AZO) films by measurement refractive index variations. It is therefore worthy of publication; however, there are some indications that must be addressed to the manuscript.
Question:
Authors say: “Unlike to SPR generation which exhibits a single resonance maximum, in LMR multiple resonances with tunable sensitivity can be generated,” Why in the experimental curves do not show multiple resonances? Authors say: “thin films with different Zn and Al metal precursors in proportions of 98:2 and 92:8 wt% were deposited onto the fiber structures,” Why do the authors choose these concentrations of Al: Zn? Are these alloys commercial? what purity? It would be advisable for the authors to make a description of instruments and materials used and described in the manuscript (Zn:Al, quartz crystal monitor, gas flowmeter, thickness gauge, etc.) Authors say: “The aforementioned structures require the use of techniques such as micromachining and chemical attack, whose processes are usually inaccurate, which acts in detriment of the precise study about the generalized characteristics of the sensor” …” The single-mode-multimode-single-mode (SMS) structure used in our experiments is formed by a multimode fiber (MMF) segment fusion spliced between two single-mode fiber,” Why do authors use the chemical attack process in their MMF if it is inaccurate? Author say: “As it was demonstrated in the theoretical model proposed by the group at the Public University of Navarra [7, 33, 34], the number of reflections at the fiber-coating interface as a function of the angle of incidence N(θ)=L/dtan(θ) (neglecting the skewness angle), depends on the length (L) of the coated region and the fiber diameter (d),” …. “Then, in order to obtain a field distribution at the end of the MMF as an image of the input field, the length of the MMF is selected,” Do the authors use certain values in other works or simulate the sensor response? Authors say: “At the same time the MMI waveband approaches to reach a wavelength region in which the self-image effect fades [32] which is caused by the LMR phenomenon [33]” What does MMI mean? Authors say: “Zn:Al alloys were used: one target of Zn:Al, 92%:8% and another one of Zn:Al, 98%:2% atomic concentration. Optical fibers were mounted on glass substrates,” Is the thin film coating only made on one side, every 180 degrees? or on the entire diameter of the fiber? Authors say: Deposition rate was monitored with a quartz crystal sensor interfaced with an external computer. What do you use the quartz crystal monitor for? to monitor the thickness and evaporation rate of the thin film? Then, so that they measure the thickness of the thin film with a filmetric equipment? Why don't the authors make a brief description of the homemade NCF?Author Response
In this article the authors claim in an optical fiber sensors using an optical-fiber structures based on multimode fibers and NCF coated with an aluminum-doped zinc oxide (AZO) films by measurement refractive index variations. It is therefore worthy of publication; however, there are some indications that must be addressed to the manuscript.
Question:
Authors say: “Unlike to SPR generation which exhibits a single resonance maximum, in LMR multiple resonances with tunable sensitivity can be generated,” Why in the experimental curves do not show multiple resonances?
Author’s response:
Thank you, we appreciate all your comments and the time you expended reviewing our manuscript. Concerning this question, our study is focused on the application of the LMR phenomenon observed in AZO coated fiber structures as a refractive index sensor but specifically for the 1.55 µm wavelength region, since it is a highly attractive waveband for fiber optic systems. In addition, our input signal for the analysis of the sensor response in the region of 1.55 µm was a fiber coupled LED source with stable low noise emission from 1400 to 1700 nm. The LED source simplified our experimental analysis but limited it to a short wavelength range. In this sense, as a first instance we do not considered to show the resonances related with the LMR orders exhibited in the visible wavelength region (3rd and 4th orders). In order to clarify this concern, we measured the transmitted spectrum of the fiber structure based on a MMF. The figure is attached below (see the figure in the PDF version of the response). The measurement was performed with a white light source based on a dichroic lamp as input signal. The light emitted was introduced into the fiber through a lenses setup. Although the light emitted by the lamp has a significantly noisy spectrum that ranges from 400 to 1000 nm, in the output transmitted spectrum of the figure it can be observed two LMR orders shown as transmission notches: The 4th order in ~ 430 -520 nm and the 3rd order at ~ 570-780 nm.
Authors say: “thin films with different Zn and Al metal precursors in proportions of 98:2 and 92:8 wt% were deposited onto the fiber structures,” Why do the authors choose these concentrations of Al: Zn? Are these alloys commercial? what purity? It would be advisable for the authors to make a description of instruments and materials used and described in the manuscript (Zn:Al, quartz crystal monitor, gas flowmeter, thickness gauge, etc.)
Author’s response:
We agree with the reviewer that more details about films’ preparation must be included. For this reason, we have included the purity of targets, together with the company that have been bought. Also we have included an explanation of why we choose 2% and 8% as Al-doping. For AZO films, Al-content must be between 2% and 8%, being 1% to 2% the most reported as an almost-metallic film. Also in lines 180 to 184, we have added the purity of gases (Ar and O) and also we have made clear that each mass flowmeter was calibrated for the specific gas that pass across it. This last statement can be found included in Section 3.
Authors say: “The aforementioned structures require the use of techniques such as micromachining and chemical attack, whose processes are usually inaccurate, which acts in detriment of the precise study about the generalized characteristics of the sensor” …” The single-mode-multimode-single-mode (SMS) structure used in our experiments is formed by a multimode fiber (MMF) segment fusion spliced between two single-mode fiber,” Why do authors use the chemical attack process in their MMF if it is inaccurate?
Author’s response:
The use of fiber structures based on MMF has been reported before as a common method for the implementation of SPR and LMR fiber sensors. In this sense, different authors have used MMF with plastic or silica cladding. Then, in order to improve the performance of the SPR/LMR sensor, it is necessary to remove the cladding by using different methods of chemical attack or micro-machining. In our manuscript we wanted to highlight the use of a No-core fiber (NCF) which does not require a cladding removing process. Therefore, in addition to the fiber structure based on NCF, we have also implemented MMF structures with cladding removed by chemical attack by way of comparison in sensor performance. As it is shown in the obtained results (Fig. 6), the use of NCF-based structures improves the sensitivity of the sensor, additionally its simplicity and repeatability in the construction process.
Author say: “As it was demonstrated in the theoretical model proposed by the group at the Public University of Navarra [7, 33, 34], the number of reflections at the fiber-coating interface as a function of the angle of incidence N(θ)=L/dtan(θ) (neglecting the skewness angle), depends on the length (L) of the coated region and the fiber diameter (d),” …. “Then, in order to obtain a field distribution at the end of the MMF as an image of the input field, the length of the MMF is selected,” Do the authors use certain values in other works or simulate the sensor response?
Author’s response:
The authors use a mathematical model that describes the propagation of light through material-coated optical fibers in an optical transmission arrangement. For example, in ref. [34] the model was transferred to ITO films that behave similarly to AZO by behaving as transition metal oxides where the dielectric constant is modeled with the Drude model for the material. From the model they performed a simulation incorporating some parameters which were obtained experimentally.
Authors say: “At the same time the MMI waveband approaches to reach a wavelength region in which the self-image effect fades [32] which is caused by the LMR phenomenon [33]” What does MMI mean?
Author’s response:
Thanks for the observation. We had not noticed that the acronym MMI (multimodal interference) had not been defined. In the revised version of the manuscript this error has been corrected.
Authors say: “Zn:Al alloys were used: one target of Zn:Al, 92%:8% and another one of Zn:Al, 98%:2% atomic concentration. Optical fibers were mounted on glass substrates,” Is the thin film coating only made on one side, every 180 degrees? or on the entire diameter of the fiber?
Author’s response:
The AZO films covered all the external part of the fiber, leaving untouched the internal diameter. This information that was not mentioned in the paper, has already been added and can be found from line 217.
Authors say: Deposition rate was monitored with a quartz crystal sensor interfaced with an external computer. What do you use the quartz crystal monitor for? to monitor the thickness and evaporation rate of the thin film? Then, so that they measure the thickness of the thin film with a filmetric equipment?
Author’s response:
A quartz monitor can measure both evaporation rate and thickness as a function of deposition time. However, as targets erode with time, thickness measured straightforward from monitor can be inaccurate. For this reason, thickness of films was measured after deposition with a filmetrics equipment. That explanation has been included, and can be found in Section 3 from line 231.
Why don't the authors make a brief description of the homemade NCF?
Author’s response:
We appreciate your comments. A description of the NCF used for our investigation is has been added in the section 2 of the revised version of our manuscript from line 131.

Reviewer 2 Report
I am really concerned about the novelty of this work because there is another work about the same topic that can be found In sensors and actuators B from A. OZCARIZ titled AZO coated optical fiber LMR refractometers - an experimental demonstration and also a tgeoretical work from Paliwal on the same topic.
The authors should first clarify the novelty of this work compared to previous ones before going into review.
Author Response
Comments and Suggestions for Authors
I am really concerned about the novelty of this work because there is another work about the same topic that can be found In sensors and actuators B from A. OZCARIZ titled AZO coated optical fiber LMR refractometers - an experimental demonstration and also a tgeoretical work from Paliwal on the same topic.
The authors should first clarify the novelty of this work compared to previous ones before going into review.
Author’s response:
Thanks for your comment, we appreciate the time you expended reviewing our manuscript. We would like to mention the article you make reference (by Ozcariz et al.) is included in our manuscript in ref. [26], since optical properties of AZO were previously reported for its use in the development of LMR-based refraction index sensors and saturable absorbers (Ref. [27]). In this sense, we want to highlight the differences between the reported paper and the submitted manuscript as well as the novelty and significance of our work:
Since the electrical properties and the optical response of the coating material significantly vary depending on the deposition process and the material composition (in our case, aluminum doping concentration) and the fiber structure used as substrate, we have performed a detailed analysis about the optical and electrical properties of AZO films covering the fiber and studied the differences in performance by considering the Al-doping and different multimode fiber structures. XPS, UV-Visible spectroscopy and Filmetric measurements were the tools we used to perform our research, which were not reported in the referenced paper. It was also of our interest to make a contribution in providing details about the growing technique (reactive magnetron sputtering), along with the experimental parameters employed for films deposition. The proposed work is oriented to generate a reasonably useful LMR order within the 1.55 µm wavelength range, because is the most reliable waveband for optical fiber systems, which allows applications with fiber compatibility in most of the current commercial fiber systems. In case of the previously reported paper, a functional lossy-mode resonance for its use the 1.55 µm was not obtained. In addition, the prediction presented to reach this wavelength range is not favorable, since a severe LMR order widening avoids its use to develop a refractive index sensor. In that sense, this “readable” lossy-mode resonance at the C-band was already achieved in our proposed work. Based in our results, we considered that the deposition process and the characteristics of the deposited thin film play a main role for the wavelength-tuning of the generated LMR orders; then, in the proposed work we present detailed information of the AZO film deposition process we used, to obtain the desired results as well as a detailed study of the electrical, optical and composition characteristics of the AZO films deposited onto the optical fiber. It is worth to note that this information is not available in the previously reported paper, which can difficult the study of a relation between the deposition characteristics and the obtained LMR generation. Moreover, we investigated the influence of the optical fiber structure and the aluminum doping concentration in the sensitivity of the refractive index sensor as a function of the LMR order wavelength displacement. In this regard, we investigate two different optical fiber structures, one (our highlighted proposition) based on a homemade no-core fiber and the other (more similar to the multimode fiber with plastic cover investigated before for AZO-based fiber refractometer of ref. [26]) based on a commercial multimode fiber whose silica cladding was removed by chemical attack. Additionally, depositions with two different AZO targets were used: with 2% and 8% aluminum doping. The obtained results were very illustrative to optimize the operation characteristics of the proposed sensor proving the increase of sensitivity, ease of fabrication and repeatability of the proposed fiber structure based on a no-core fiber as coated sensing element. In our reported results, we have obtained a 2nd LMR order sensitivity of about 1214.7 nm/RIU, compared with the one of ref [26] of 1153.6 nm/RIU.
The manuscript was restructured taking into account your comment and the comments of the other reviewers as fundamental guidelines in order to clarify the concerns and demonstrate the significance of the proposed investigation including a modification in the Introduction Section in order to highlight our contribution.

Reviewer 3 Report
This paper presents an experimental study and demonstrates the generation of lossy-mode resonance (LMR) in optical-fiber structures based on multimode fibers coated with an aluminum-doped zinc oxide (AZO) film.
Although AZO film optical fiber sensor have been previously reported and employed in literature, the novelty and ORIGINALITY of this paper is the sensing device constructed by AZO film deposition on a single-mode-multimode-single-mode structure.
The introduction is well written. First, the introduction mentions the three main classes of optical sensing with thin-film deposited waveguides: SPR, LMR and LRSEP. Then, a SPR and LMR are presented comparatively, mentioning both advantages and disadvantages of each. This follows to the motivation for choosing the LMR technique for this paper. Next, films are discussed and the choice for AZO thin-films is motivated.
A mathematical description of the LMR technique to express the phenomena involved, e.g. attenuated total reflection, geometry dependency, transmission notch vs. incident radiation source, tenability of the resonance frequency, etc. would consistently improve this presentation. Although the authors refer to citations [7, 33, 34] later on line 139 referencing the model proposed by the research group of the Public University of Navarra, I would suggest placing such a description as a stand-alone section, or in the introduction where the LMR is described.
The description of the optical sensor given in section 2 is very good and well written. Nevertheless, the mathematical apparatus is somewhat weak.
Section 3 is also very well written, the procedure of film deposition is well detailed and the characterization results reported in figures 2 and 3 sustain the claims issued in the introduction.
Finally, the test scenario in section 4 is well described, and the measurement results confirm the claims issued in the introduction. Accordingly, the proposed optical sensor was tested and its functionality was demonstrated in laboratory environment.
Minor language and grammar revision is required. For example, in the first sentence form the abstract “multimode fibers coated with an aluminum-doped zinc oxide (AZO) films“: film should be singular. Such minor inaccuracies can be found along the paper.
Author Response
Comments and Suggestions for Authors
This paper presents an experimental study and demonstrates the generation of lossy-mode resonance (LMR) in optical-fiber structures based on multimode fibers coated with an aluminum-doped zinc oxide (AZO) film.
Although AZO film optical fiber sensor have been previously reported and employed in literature, the novelty and ORIGINALITY of this paper is the sensing device constructed by AZO film deposition on a single-mode-multimode-single-mode structure.
The introduction is well written. First, the introduction mentions the three main classes of optical sensing with thin-film deposited waveguides: SPR, LMR and LRSEP. Then, a SPR and LMR are presented comparatively, mentioning both advantages and disadvantages of each. This follows to the motivation for choosing the LMR technique for this paper. Next, films are discussed and the choice for AZO thin-films is motivated.
A mathematical description of the LMR technique to express the phenomena involved, e.g. attenuated total reflection, geometry dependency, transmission notch vs. incident radiation source, tenability of the resonance frequency, etc. would consistently improve this presentation. Although the authors refer to citations [7, 33, 34] later on line 139 referencing the model proposed by the research group of the Public University of Navarra, I would suggest placing such a description as a stand-alone section, or in the introduction where the LMR is described. The description of the optical sensor given in section 2 is very good and well written. Nevertheless, the mathematical apparatus is somewhat weak.
Author’s response:
We appreciate all your comments and suggestions. Regarding the mathematical description you suggest, we have restructured the section 2, including a stand-alone subsection in order to improve the description of the LMR phenomenon in optical fibers. Part of the referenced model was included in a more extended mathematical apparatus in the subsection “2.2. LMR generation in thin-film coated optical fibers”
Section 3 is also very well written, the procedure of film deposition is well detailed and the characterization results reported in figures 2 and 3 sustain the claims issued in the introduction.
Finally, the test scenario in section 4 is well described, and the measurement results confirm the claims issued in the introduction. Accordingly, the proposed optical sensor was tested and its functionality was demonstrated in laboratory environment.
Minor language and grammar revision is required. For example, in the first sentence form the abstract “multimode fibers coated with an aluminum-doped zinc oxide (AZO) films“: film should be singular. Such minor inaccuracies can be found along the paper.
Author’s response:
In this last paragraph we again want to thank all your wise comments and detailed analysis of our manuscript. In regard to language errors, we have reviewed the revised version of our manuscript to minimize these writing issues.

Round 2
Reviewer 2 Report
I thank the authors the detailed explanation of the main and significant content of their paper compared with other works on LMRs based on metal oxide coatings and AZO coatings particularly.
The explanation provided by the authors is clear and the work is presented as an extension of previous works from other groups so it should be also more clearly indicated in this paper and emphasized in the introduction.
Since this work relies on the LMR phenomenon in optical fiber it should be fair to mention and give more credit to the authors that first demonstrated experimentally the phenomenon in literature. From my point of view it should be fine to cite them in the text.
Author Response
Comments and Suggestions for Authors
I thank the authors the detailed explanation of the main and significant content of their paper compared with other works on LMRs based on metal oxide coatings and AZO coatings particularly.
The explanation provided by the authors is clear and the work is presented as an extension of previous works from other groups so it should be also more clearly indicated in this paper and emphasized in the introduction.
Since this work relies on the LMR phenomenon in optical fiber it should be fair to mention and give more credit to the authors that first demonstrated experimentally the phenomenon in literature. From my point of view it should be fine to cite them in the text.
Author’s response:
We appreciate all your comments, suggestions and the time you expended reviewing our manuscript. In this regard, we agree with your suggestion to include more information by giving credit to the precursor research groups that have motivated our proposed work. In this sense, we have included information in the introduction section in order to emphasize the previous work of these groups, the credit they deserve, and the importance of them as the base of our proposed work.
